# Melatonin and Myo-Inositol: Supporting Reproduction from the Oocyte to Birth

**DOI:** 10.3390/ijms22168433

**Published:** 2021-08-05

**Authors:** Michele Russo, Gianpiero Forte, Mario Montanino Oliva, Antonio Simone Laganà, Vittorio Unfer

**Affiliations:** 1R&D Department, Lo.Li. Pharma, 00156 Rome, Italy; m.russo@lolipharma.it (M.R.); g.forte@lolipharma.it (G.F.); 2The Experts Group on Inositol in Basic and Clinical Research (EGOI), 00161 Rome, Italy; dr.montanino@gmail.com (M.M.O.); antoniosimone.lagana@uninsubria.it (A.S.L.); 3Department of Obstetrics and Gynecology, Santo Spirito Hospital, 00193 Rome, Italy; 4Department of Obstetrics and Gynecology, “Filippo Del Ponte” Hospital, University of Insubria, 21100 Varese, Italy; 5System Biology Group Lab, 00161 Rome, Italy

**Keywords:** melatonin, myo-inositol, fertility, pregnancy, dietary supplementation, gestation outcomes, ART procedures

## Abstract

Human pregnancy is a sequence of events finely tuned by several molecular interactions that come with a new birth. The precise interlocking of these events affecting the reproductive system guarantees safe embryo formation and fetal development. In this scenario, melatonin and myo-inositol seem to be pivotal not only in the physiology of the reproduction process, but also in the promotion of positive gestational outcomes. Evidence demonstrates that melatonin, beyond the role of circadian rhythm management, is a key controller of human reproductive functions. Similarly, as the most representative member of the inositol’s family, myo-inositol is essential in ensuring correct advancing of reproductive cellular events. The molecular crosstalk mediated by these two species is directly regulated by their availability in the human body. To date, biological implications of unbalanced amounts of melatonin and myo-inositol in each pregnancy step are growing the idea that these molecules actively contribute to reduce negative outcomes and improve the fertilization rate. Clinical data suggest that melatonin and myo-inositol may constitute an optimal dietary supplementation to sustain safe human gestation and a new potential way to prevent pregnancy-associated pathologies.

## 1. Introduction

### 1.1. Background

Human reproduction is a complex process in which several molecules contribute to the correct development and birth of the offspring [1]. The progression of gestation may be affected by molecular dysfunctions, lowering the chances of a physiological pregnancy [2]. Among the different molecules involved in the pregnancy, melatonin and myo-inositol (myo-ins) actively sustain the reproduction process and favor positive gestation outcomes [3]. An increasing amount of data are strengthening the idea that these species indicate higher quality of oocyte and embryo [4]. Several clinical studies have claimed that ensuring a proper amount of melatonin and myo-ins in women willing pregnancy improves the reproductive process and reduces the negative events [5,6].

The aim of this review is to describe the importance of dietary supplementation with melatonin and myo-ins, describing their key role in the regulation of the steps characterizing the conception of a new life. This review also provides an overview on the implications of these two important molecules on the reproduction process, from oocyte to birth.

### 1.2. Melatonin

Melatonin or N-acetyl-5-methoxytryptamin is an indoleamine characterized by two functional groups that are fundamental for binding the specific receptors and for amphiphilicity, allowing the entrance in any cell, compartment, or body fluid [7]. Melatonin is a small neuro-hormone released by the pineal gland (or epiphysis), detectable in all vertebrates. The circadian fluctuations of its secretion play a crucial role in the coordination of the sleep-wake cycle with circadian and seasonal rhythms [8].

In humans, as in a greater part of mammals, the circadian clock is made up by an organized and hierarchical structure, which presents a top input mediator known as the suprachiasmatic nucleus (SCN) localized inside the hypothalamus [9]. The SCN regulates the circadian physiology and behavior, including sleep patterns, body temperature, hunger, neuroendocrine and autonomic effects, translating the environmental light–dark cycle into an internal temporal order [10]. During darkness melatonin is secreted by the pineal gland in a circadian manner and released into the blood vessels. However, it is also produced in a non-cyclic way in the mitochondria of all type of cells [11].

In 1958 Lerner and colleagues firstly isolated melatonin [12], which was further identified as a factor involved in reproductive physiology. Further research highlighted that the altered pubertal development directly associates to the onset of a human pineal gland tumor, allowing the hypothesis that some pineal factor may influence the reproductive system [13]. A significant step in this direction was the observation that abolition of plasma melatonin rhythmicity is related to disruption of the neural connection between the SCN and the pineal gland. In addition, the circadian clock genes regulation allowed by melatonin occurs in several tissues of the reproductive axis, both in the embryo and in the adult [14].

A functional connection between melatonin and the pituitary-gonadal axis emerged after the commercial availability of melatonin, confirming that the pineal gland is involved in the regulation of reproductive functions particularly in seasonal breeders [15]. Melatonin is involved in the whole reproduction process [16], revealing antioxidant functions [17], hormonal regulation [18], and improving gamete functionality [19].

### 1.3. Myo-Inositol

On the other hand, myo-ins is chemically identified as a cyclohexane with six hydroxyl groups, isolated for the first time by Scherer in 1850 from muscle extracts and named myo-inositol from the Greek word “mios”, meaning muscle [20]. Inositols refers to a family of 9 stereoisomers, among which myo-ins is the most diffused in nature, including in mammals [21]. Myo-ins can either derive from the diet, as it is principally found in corn, nuts, and fruits [22], or it can be synthetized from D-glucose in the human body by various tissues, including brain, liver, kidney, mammary gland, and testis [23].

Different authors agree that myo-ins plays a pivotal role in cellular metabolism. Its presence in the human body is necessary to biosynthesize several molecular components of the biochemical machinery [24]. One of the main impacts of myo-ins on the human physiology is the regulation of several molecular pathways directly connected with the reproductive functions. Adequate quantities of myo-ins ensure a proper functioning of the male and female reproductive cells; in fact, impaired or deregulated concentrations in follicular fluid (FF) or semen are frequently associated with pathological features and reduced fertility [25]. In addition, myo-ins supplementation likely exerts an improvement in several pathological conditions, like polycystic ovarian syndrome (PCOS), promoting the restoration of physiological functioning and increasing fertility in the affected patients. Moreover, it seems to have a positive impact in assisted reproduction technologies (ART) and in vitro fertilization (IVF) [26].

## 2. Improvement of the Ovulation Quality—Oogenesis

Melatonin is a key regulator of several physiological functions including the circadian rhythm, which aligns the metabolism machinery with a 24-h daily cycle. A cluster of “clock genes” expressed in most of tissues, guarantees such function [27]. Interestingly, melatonin can modulate molecular mechanisms of the female reproductive apparatus by sending photoperiod-dependent signals that regulate ovarian functions, follicular development process and ovulation [16].

Melatonin also plays a fundamental role in achieving good oocyte quality and in supporting a physiological follicular phase, mainly due to its free-radical-scavenging properties [28]. Melatonin, in fact, controls reactive oxygen species (ROS) and reactive nitrogen species (RNS) produced by cellular metabolism. In detail, it quenches both ROS and RNS, including superoxide anion (O_2_^-^), hydroxyl radical (OH), singlet oxygen (^1^O_2_), hydrogen peroxide (H_2_O_2_), hypochlorous acid (HOCl), nitric oxide (NO), and the peroxynitrite anion (ONOO^-^) [29]. Interestingly, melatonin also stimulates several enzymes that metabolize potentially reactive species or that induce the endogenous synthesis of superoxide dismutase (SOD), glutathione peroxidase (GPx) and glutathione reductase, enzymes with antioxidant activities [28]. Unlike other antioxidants, melatonin reacts with free radicals in a “scavenging cascade” producing stable end-products with antioxidant properties. Moreover, it cooperates with other non-enzymatic antioxidants and induces the synthesis of antioxidant enzymes by binding to cytosolic or nuclear binding sites on specific membrane receptors [30,31].

Numerous studies pointed out that oxidative stress in the female reproductive apparatus is the main cause of ovarian aging, with a negative impact on fertility [6]. Melatonin has specific receptors in the ovaries, indicating its role in maintaining physiological status. Melatonin receptors 1 and 2 (MT1 and MT2), structured as 7-transmembrane receptors coupled to G proteins, are present in human granulosa cells, luteal cells and in rat ovaries (antral follicles and corpus luteum) [28,32]. The role in the reproductive apparatus is further demonstrated by the evidence that melatonin content is much higher in the preovulatory FF with respect to serum levels. Additionally, increased concentration of melatonin in growing follicles is crucial to avoid atresia, as it reduces apoptosis of critical cells [33,34]. In conclusion, a positive association exists between melatonin quantity in the FF and oocyte quality, representing an important element in the development of a competent oocyte for successful fertilization [33].

Data collected from different experimental study demonstrate that the scavenger activity of melatonin allows correct oogenesis. Imbalanced ROS concentration and antioxidant activity result in reduced oocyte quality due to the alterations and damages caused by oxidative stress. Studies in mice demonstrated that ovarian stimulation with human chorionic gonadotropin (hCG), followed by stimulation with pregnant mare serum gonadotropin (PMSG), resulted in increased levels of 8-hydroxy-20-deoxyguanosine (8-OHdG), a marker of DNA damage, and of hexanoyl-lysine (HEL), a marker of lipid peroxidation [35].

A clinical trial performed by the same authors included patients with a high rate of denatured oocytes (poor oocyte quality) subjected to IVF, revealing high levels of 8-OHdG in the FF. Such elevated concentration of 8-OHdG seemed associated with low fertilization rates, indicating that oxidative stress during the ovulatory process reduces the oocyte quality and increases the risk of infertility [36]. Most interestingly, the authors observed a significant inverse correlation between melatonin concentration and the presence of 8-OHdG in the FF. Furthermore, patients who received melatonin supplementation (3 mg/day, taken at 10 p.m.), exhibit increased concentration of melatonin in the FF and strongly decreased levels of 8-OHdG. These data underline the crucial role of melatonin in reducing the oxidative stress in ovarian follicles, protecting oocytes and granulosa cells, and thus positively contributing to female fertility (Figure 1) [36].

Melatonin, hence, participates in various cellular processes and has a prominent role in reproductive physiology. Analogously, myo-ins is an organic compound with a crucial biological role in the entire human organism and it is heavily involved in regulating the reproductive functions. Myo-ins has a role in human reproduction from the very early phases of folliculogenesis, participating in the selection of the dominant follicle during the oogenesis process [37]. It acts as second messenger in the follicle stimulating hormone (FSH) signaling, activating the aromatase enzyme to increase the production of estradiol (E2), which is directly involved in the selection of the dominant follicle that will undergo oogenesis [37]. Myo-ins is an important regulator of follicular microenvironment, contributing to oocyte development. In women undergoing IVF techniques, higher concentrations of myo-ins were observed in the FFs of women with better-quality oocytes (Figure 1) [38]. Supplementation of myo-ins results in improved meiotic progression of mouse germinal vesicle oocytes, enhancing intracellular Ca^2+^ oscillation [39].

A wealth of information regarding the importance of myo-ins in the ovaries comes from studies in pathological scenarios that lead to the need for assisted fertilization protocols. In particular, evidence derives from studies on patients with PCOS, the most common endocrinological syndrome in women of reproductive age, which leads to lower quality and maturity of oocytes and to consequently lower quality of embryos [40]. PCOS is often characterized by alterations of the inositol profile, both at ovarian and at global levels [41]. Therefore myo-ins supplementation is useful in the treatment of PCOS, as it is an important constituent of follicular microenvironment, mediates FSH signaling and promotes oocyte development and maturation [42].

The benefits of myo-ins administration in this pathological context emerge from a recent study carried out on a cohort of 50 infertile PCOS women candidates for the IVF treatment. The participants, aged 20–35 years and with partner’s normal semen analysis, were randomly divided in two groups: the study group received a daily dose of 4 g myo-ins, combined with 400 mcg folic acid (FA); the control group received only 400 mcg FA. Treatment started one month before ovum pick up in both groups [43]. The results demonstrated that the percentage of metaphase II (MII) oocytes, named mature oocytes, and fertilization rate significantly increased in the study group compared with controls. In addition, the authors reported that the embryo quality improved after treatment with myo-ins. Furthermore, the study group exhibited a higher gene expression of phosphoglycerate kinase 1 (PGK1), regulator of G-protein signaling 2 (RGS2) and cell division control protein 42 homolog (CDC42), considered marker genes of oocyte quality in granulosa cells [43].

If the beneficial role of myo-ins is extensively documented in the PCOS context, it is likewise established that myo-ins supplementation has a positive effect also in non-PCOS women undergoing IVF. In particular, in the trials of Lisi et al. and Caprio et al., non-PCOS women were divided in two group and treated with six-day stimulation and three-month supplementation as follows: the study group received 200 mcg/die folate and 2000 mg/die myo-ins twice a day; the control group received 400 mcg/die folate (for 3 months). The results demonstrated that the women treated with myo-ins needed fewer FSH quantities to reach follicular maturation, i.e., a total dose of 2.084 IU vs. 2.479 IU in the control group [44]. Here too, myo-ins administration promotes a higher rate of MII oocytes (80.5% myo-ins vs. 66.6% control) and in addition improves the ovarian sensitivity index (1.88 vs. 1.54) [45].

Fueled by recent studies, the role of dietary supplementation with myo-ins and melatonin in supporting fertility is achieving growing recognition. In particular, new evidence highlighted that the simultaneous treatment with both molecules effectively improves the quality of oocytes and embryos, increases clinical pregnancy rate even after previous IVF failures, and results in a better implantation rate [46,47].

A randomized controlled trial evaluated the synergistic effect of a simultaneous administration of melatonin and myo-ins on oocyte quality and fertilization rate [4]. Five-hundred and twenty-six PCOS women, elected for the first IVF treatment, were divided in three groups: control (FA: 400 mcg), group A (myo-ins 2000 mg and FA 200 mcg, twice a day, plus melatonin 3 mg/die), and group B (myo-ins 2000 mg and FA 200 mcg, twice a day). The patients were treated from the first day of the cycle until 14 days after embryo transfer. The results indicated that intrafollicular concentration of melatonin was significantly higher in group A (213 ± 35 pg/mL) vs. group B (69 ± 23 pg/mL) and vs. control group (54 ± 3.5 pg/mL). Differences were also found in the IU of gonadotropin required (group A: 2058 ± 233 vs. group B: 3113 ± 345 and vs. control group: 3657 ± 633). Furthermore, an increased number of mature oocytes were obtained from group A with respect to group B and control group. In addition, the percentage of grade I embryos increased in group A (45.7% vs. 30.4% in group B and 25.6% in the control group) [4]. Taken together, these aspects not only indicate the involvement of melatonin and myo-ins from the early gestational phases, but also sustain the positive impact of a dietary supplementation with these molecules in women that seek pregnancy.

## 3. Oocyte Maturation and Fertilization

Melatonin is necessary for good-quality oocytes and is a key factor to start the cascade of physiological events that brings a new life. However, its role seems crucially important in each step of the reproductive process, also influencing the fertilization stage. Elevated quantities of melatonin in the FF directly relate with good pregnancy rate (Figure 1). Indeed, melatonin concentration is higher in large follicles (diameter ≥ 18 mm) than in intermediate (10–17 mm) and in small follicles (<10 mm) [34]. In parallel, the abundance of oxidative stress markers in the FF seems to be inversely related to the amount of melatonin, suggesting its decisive role in counteracting the oxidative stress in follicular environment and thus in preventing pregnancy failure.

A clinical trial assessed the positive effects of melatonin administration in patients who previously failed to conceive with intra-uterine insemination technique. The participants were divided in three groups: group A (control), group B (medium dose of melatonin), and group C (elevated dosage of melatonin). The authors demonstrated that group A exhibited lower FF melatonin levels and strongly decreased IVF outcome parameters such as fertilization rate, number of mature oocytes and high-quality day 3 embryos, when compared to the other groups. Moreover, patients in group A had higher levels of oxidative stress markers (ROS, TBARS, and 8-OHdG). On the contrary, group B and group C had decreased oxidative status. The authors observed the same trend for IVF outcome parameters, which were better in group B compared to group A, and achieved a maximum in group C. The high melatonin levels in the FF of group C positively correlate with the improved number of MII oocytes, normal fertilized oocytes, early cleaved zygotes, blastomeres with regular symmetry, and high-quality day 3 embryos [48].

Interestingly, the oxidative stress is a direct cause of DNA damage, which modulates the expression of apoptosis-associated microRNAs (miRNAs) in the granulosa cells of developing oocytes. On the other hand, the administration of melatonin upregulates the expression of several miRNAs with a crucial role in oocyte maturation and embryo quality. Furthermore, melatonin reduces the cell-free DNA (cfDNA), a useful marker of DNA breakage caused by oxidative stress [48]. Melatonin significantly ameliorates the maturation of bovine oocytes by regulating the correct cytoplasmic distribution of organelles, increasing intracellular levels of glutathione (GSH) and adenosine triphosphate (ATP), enhancing antioxidant gene expression levels, and enhancing fertilization-related events. All these effects resulted in a better fertilization rate and developmental ability and safety [49].

As mentioned, data obtained in the recent years, especially from ART practice, contributed to strengthen the idea of melatonin as a promoter of higher fertilization rate. Melatonin, indeed, also seems to improve the function of the in-vitro maturation medium (IVM) used for intra-cytoplasmatic sperm injection (ICSI) procedures. Among the different adverse effects on cellular environment, the oxidative stress causes lipid oxidation and stiffening of the plasma membrane. Melatonin increases clathrin-mediated endocytosis (CME), which restores membrane plasticity and induces cyclic adenosine monophosphate (cAMP) downregulation, thus favoring the progression of MII oocytes and promoting both fertilization and early embryo development [50].

Like melatonin, also myo-ins is nowadays considered a pivotal molecule that improves the fertilization rate (Figure 1). In female mammals, myo-ins is more abundant in the reproductive organs than in blood. This supports a different function covered in the ovaries, where it acts as a second messenger of FSH, and improves oocyte and embryo quality. Myo-ins, indeed, promotes the maturation of oocytes, which possess inositol 1,4,5-trisphosphate (IP3) receptors and myo-ins transporters, just like maturing embryos [51]. In mouse models, myo-ins induces the meiotic progression of oocytes into fertilization-competent eggs, and oocyte maturation failure can be frequently observed when ovarian levels of myo-ins are low. Additionally, myo-ins provides a suitable environment for the gametes by facilitating oocyte transport in the oviduct and enhancing spermatozoa motility, overall increasing the fertilization chances [51].

Data retrieved from an in-vitro study on mouse embryos preimplantation demonstrate that myo-ins administration enhances the progression to advanced developmental stage, favoring the development to expanded blastocyst stage and increasing the percentage of blastomeres forming the embryos at the blastocyst stage [52]. The underlying molecular mechanism involves phosphatidylinositol (PtdIns), in particular, phosphatidylinositol (3,4,5)-trisphosphate (PIP3), which is synthetized by phosphatidylinositol-3-kinase (PI3K). The enzyme phospholipase C (PLC) converts PIP3 into diacylglycerol and IP3, which in turn promotes cell proliferation by mobilizing Ca^2+^ storages. This signaling also activates the protein kinase B (PKB/Akt) pathway, which is known to enhance the proliferation of mouse embryo blastomeres, possibly explaining the increased synthesis of PtdIns and the resulting increase in PIP3 level observed. Moreover, the induction of Akt phosphorylation likely enhances the development rate of cultured embryos [53].

The role of myo-ins in this crucial step of gestation is also well documented in humans by several authors. Data suggest that, when used as dietary supplement, myo-ins is effective in ameliorating ART outcomes [54]. In a systematic literature review and meta-analysis, Zheng and colleagues demonstrate the importance of myo-ins administration to infertile non-PCOS patients undergoing ovulation induction for ICSI or IVF-ET [51]. A total of 935 women, from 7 clinical trials were included. The results confirm the efficacy of myo-ins administration in achieving grade 1 embryos and in improving the clinical pregnancy rate, with reduced abortion rate. The induction of the follicular maturation in the treated group required significantly lower IU of gonadotropins, such as recombinant FSH (rFSH), than the control group. These findings highlight a positive impact of myo-ins on clinical pregnancy rate in infertile women undergoing ART procedures, increasing the number of suitable oocytes with lower amounts of drugs required for the stimulation [55].

The same beneficial effect was observed in PCOS patients. Indeed, in euglycemic PCOS patients undergoing ovarian stimulation for ICSI, treatment with myo-ins significantly increased the number of mature oocytes, good-quality embryos, and total pregnancies in comparison to other treatments [56]. In another study on PCOS women undergoing IVF, the treatment with myo-ins for 12 weeks reduced the number of degenerated oocytes, promoted a better fertilization rate, increased the number of transferred embryos, and improved the embryo quality [52]. Such evidence leads to a remarkable conclusion: myo-ins supplementation exerts an important positive effect on the oocyte health and consequent development, supporting such intervention in ART techniques.

In conclusion a growing number of data and study demonstrates the positive effect of the supplementation with melatonin and myo-ins in the reproduction process. The benefits deriving from these treatments are clearly evidenced in PCOS women subjected to IVF procedure. Both melatonin and myo-ins promote pregnancy by improving the number of grade I embryos (good quality) and the fertilization rate [4]. Noteworthy, supplementation with these species improves oocytes and embryo quality during IVF in women who had failed to conceive in a previous cycle, increasing the possibility of a physiological gestation [31,46].

## 4. Blastocyst Development and Implantation

Interestingly, the role of melatonin and myo-ins is not only limited to the early phases of a newborn creation, but recent findings highlighted their involvement in blastocyst development and implantation (Figure 1), as in embryo development. It seems that there is a “window” of uterine receptivity, defined as a limited period when the receptive state of the uterus is synchronized with the activated state of the blastocyst, sustaining attachment. To achieve a correct implantation, evidence demonstrates that a fine spatiotemporal tuning between various growth factors, cytokines, lipid mediators, and transcription factors mediated by steroid hormones, is essential for uterine preparation [57].

Several of those molecules are directly regulated by melatonin, in a crosstalk mediated by its membrane receptors MT1 and MT2. Melatonin stimulates the expression of MT1, MT2, protein p53 and leukemia inhibitory factor (LIF), activating the signaling pathways involved in embryo implantation [58]. At molecular level MT2 receptor is more active than MT1 in the reproduction process and it is upregulated. The increased expression of MT2 promotes the activation of protein p21. This, in turn, induces protein p38-mediated phosphorylation of p53, which modulates LIF expression thus promoting the embryo implantation [6,59].

In-vitro treatment of mouse embryo at 2-cell stage with melatonin guarantees improved percentage of morula formation, enhances the development rate of blastocysts and hatching blastocysts. Melatonin also induces a significant increase of the total cell number (TCN), trophoectoderm (TE), and inner cell mass (ICM) of the blastocysts. Moreover, the authors observed a better incidence of implantation for the transferred embryos, compared to the control group, suggesting that melatonin is involved in growth metabolism and positively affects both the quality and the numbers of developing embryos [60].

In a pre-clinical study, He and colleagues demonstrate that intraperitoneal injection of 15 mg/kg melatonin in murine models increased the number of implantation sites and the litter size in the treated group. Additionally, the histological analysis of the endometrium highlights a larger endometrial area in mice treated with melatonin, with an increased density of uterine glands [61]. Uterine glands are indeed considered the primary source of nutrients for early embryos, so that their development directly affects the chances of embryo survival and implantation as well as the establishment and maintenance of gestation [62].

Authors proposed that melatonin promotes blastocyst activation and endometrial receptivity by modulating the hormone expression in the uterus [61]. On the one hand, it induces the expression of progesterone (P) and progesterone receptor A (PRA) at day 6 from fertilization, promoting endometrial luminal epithelial differentiation, stromal cell proliferation, and decidualization. On the other hand, melatonin downregulates the secretion and the activity of E2 to avoid premature uterine contractions. E2 is probably produced and secreted by the blastocyst itself around day 4, allowing the acquisition of the implantation competence, necessary to start the implantation process [63]. This phase involves several factors originating from both the blastocyst and the uterus. Melatonin-treated mice exhibit increased heparin-binding epidermal growth factor-like growth factor (HB-EGF), which has a pivotal role in embryo–uterine interactions [64]. Furthermore, HB-EGF is expressed in the luminal epithelium at day 6, and melatonin treatment promotes the activation of epidermal growth factor receptor (ErbB1-4 receptors, which also bind HB-EGF) located on the blastocyst. The ErbB1/4 receptors as well as HB-EGF are positively correlated with uterine receptivity, suggesting that melatonin improves the implantation competence of both the blastocyst and the uterus, and promotes the activation of the blastocyst in vivo [61].

In a recent clinical trial, Zhao and colleagues collected a total of 193 immature oocytes from women undergoing controlled ovarian hyperstimulation (COH) cycles. Such oocytes usually have very low chance to give birth to offspring and are discarded during ART practice. The authors divided the collected oocytes into two groups to undergo IVM, insemination and embryo culture: M-group (*n* = 105), with IVM medium containing 10 μM melatonin; NM-group (*n* = 88), with IVM medium without melatonin. With respect to the NM-group, the authors observed great improvement in the blastocyst formation rate in the M-group (28.4% vs. 2.0%), with a very low rate of aneuploidy. Subsequently, the same authors performed other ICSI procedures and transferred the cultured cleaved embryo in two patients. Both patients achieved pregnancy and gave birth to three healthy children. Apparently, melatonin in the IVM medium preserves mitochondrial oxidative phosphorylation in the oocytes through the inhibition of environmental stress, providing the required ATP for the ensuing embryo development [65].

Several studies described the correlation of melatonin signal and the connected circadian rhythm with the determination of the immune system in the developing embryo. Although the involved mechanisms are not fully elucidated, the deregulation of the immune responses often seems to cause aberrant pregnancy outcomes as recurrent pregnancy loss, implantation failure, preeclampsia, preterm birth, and intrauterine growth restriction [66]. A study analyzing the uterine transcriptome from pre-receptive to receptive stage in the human endometrium, utilizing RNA Sequencing, highlighted that novel transcripts molecularly interconnected with the “circadian rhythm” pathway, are significantly upregulated in the increasing of uterine receptivity [67]. Knockdown with siRNA of a central circadian clock protein named period circadian protein homolog 2 (Per2) in endometrial cell culture induces a greatly disorganized decidual response and alters the transcripts expression. The model proposed involves deregulated pro-inflammatory decidual response, which expands the window of endometrial receptivity, increasing the risk for out-of-phase implantation and pregnancy failure [68].

The normal implantation process is divided into three subsequent phases: apposition, attachment, and penetration. Across the entire process, myo-ins is actively internalized and quickly incorporated in the membranes, thus ensuring a robust cellular uptake that increases from the one-cell to the blastocyst stage. It seems that mammalian transition from preimplantation to implantation stage shows a parallel increase in cellular requirement of myo-ins [69]. This aspect suggests that the myo-ins addition to culture media used for IVF practice may strongly improve the positive outcomes of embryo development.

In this regard, Kuşcu and colleagues demonstrated the beneficial impact of myo-ins added to culture media of preimplantation mice embryos. Fertilized embryos cultured in the presence of myo-ins since blastocyst formation display a strong activation of PKB/Akt pathway. This finding suggests that myo-ins uptake induces PtdIns formation, increases Akt phosphorylation that promotes cell proliferation, and can be responsible for the faster developmental rate of cultured embryos [70]. Moreover, the authors observed an increase in the overall rate of live births after transfer of blastocysts developed in the presence of myo-ins, with absence of toxic effect [70].

The central role of myo-ins, suggested from the active cellular uptake and the incorporation during blastocyst development, is further supported from data obtained with bovine embryos. Myo-ins is absorbed in growing amounts from 2-cell stage to morula and blastocyst, and in parallel increasing in total amount of inositol phosphates is observed with a maximum rate at day 7. Myo-ins uptake in preimplantation cattle embryos and its incorporation into PtdIns strengthen the crucial role of myo-ins in blastocyst formation and development [71]. Further data also demonstrate that the Inositol-Phosphate pathway is directly involved in the beginning of the blastocyst elongation phase at day 14–day 16 in cattle embryos [72].

The majority of information regarding the effect of myo-ins in the development of human embryos derives from evidence on PCOS patients. Indeed, systematic reviews of randomized controlled trials pointed out the pivotal role of inositols to improve the symptoms in women with PCOS and to increase their fertility [73,74]. In particular, myo-ins supplementation strongly improves pregnancy outcomes, including enhanced oocyte follicular development and oocyte maturation. Furthermore, myo-ins contributes to ameliorate stimulation protocols and pregnancy outcomes in IVF procedures [74].

The ability of myo-ins to help achieving gestation in PCOS women derives from the improvement of the oocyte and embryo quality, usually common causes of IVF-ET failures in this type of patients [56]. Data retrieved from mouse embryo assays highlight the positive effect of the addition of myo-ins in culture media before ICSI procedures. Indeed, the zygotes cultured in the presence of myo-ins reported a larger proportion of 4-cell embryos and an increased percentage of 8- and 16-cell embryos, as well as a corresponding decrease in embryos at previous stages, with respect to controls. Moreover, myo-ins gathered a faster embryo developmental rate and a more rapid blastocyst expansion, almost doubling that of the control group [52].

In a randomized controlled trial on poor ovarian responders undergoing ICSI, 56 out of 112 included patients were treated twice a day with 2 g myo-ins combined with 200 mg FA, every day starting one month before the stimulation cycle until the hCG triggering day. The authors found that myo-ins administration strongly increased the embryo quality and the implantation rate compared to the control group treated with FA alone [75].

The active involvement of myo-ins in molecular mechanisms that allow the beginning of a new life, together with the positive correlation between melatonin amount and pregnancy progression, outline the interesting efficacy of a dietary supplementation with these molecules to improve the gestational outcomes. Noteworthy, a synergistic effect of myo-ins and melatonin was observed for safer embryo development in patients undergoing IVF procedures. The participants received either myo-ins (2000 mg twice a day) or myo-ins (2000 mg twice a day) plus melatonin (3 mg/die), from the first day of the cycle until 14 days after embryo transfer. The results demonstrate that myo-ins plus melatonin enhance oocyte and embryo quality more than myo-ins alone, suggesting that the combination can be integrated routinely in association with drugs for ovarian stimulation during IVF cycles [4].

## 5. Fetus Growth

The blastocyst implantation in the uterine wall is followed by the establishment of invasive syncytiotrophoblast (STB) derived from the trophectoderm. STB surrounds the embryo and allows the entrance in the decidualized endometrium. Around day 6/day 7, the uterine epithelial cells are locally displaced to allow the embryo penetration in the stroma. Blood and uterine secretions fill the emptiness leaved by this tissue rearrangement to supply the embryo with nutrients. The trophoblast releases human chorionic gonadotropin (hCG) and, at day 12, cells from the column cytotrophoblast (CTB) begin the formation of primary villi, which branch and acquire blood vessels and connective tissue to later form the placenta [76].

Several experimental works documented that maternal-derived tissues can provide circadian information to the fetus. Investigations on murine uterine clock and light–dark cycle during gestation established that the circadian rhythm of the mice uterus is transmitted to the fetus, which only subsequently acquires a self-sustained circadian system, independent of maternal or environmental time signals [77,78,79,80].

A proper circadian rhythm in the mother is fundamental, given that the variations due to melatonin secretion rhythm directly reflect in oscillations in the fetus. Experimental results proved that altered maternal circadian rhythm, i.e., alteration or deregulation of melatonin cycles, are associated to higher risk of psychological and behavioral problems in the offspring [16]. Clinical studies demonstrated that in addition to the pineal gland, the placenta also directly produces melatonin. MT receptors are found in CTB and STB, suggesting a local production linked to a paracrine and autocrine mechanism relative to placental homeostasis and fetal development [79].

The circadian clock genes control the placental “rhythm” at mRNA level and through translational feedback loops; incorrect functioning of these processes negatively affects placental activities and consequently local melatonin production [79,80]. Experiments carried out in mouse models indicate that melatonin administration can prevent placental malperfusion by counteracting the intrauterine inflammation-related oxidative stress. Indeed, intraperitoneal injection of 10 mg/kg melatonin, 30 min before lipopolysaccharide (LPS) administration to induce inflammation, avoids placental hypercoagulation, increases the blood flow in uterine and umbilical arteries, and counteracts pro-inflammatory and oxidative environment. Most of all, the treatment protects the fetus from ventricular dysfunctions induced by LPS oxidative stress [81]. Hence, melatonin proved to protect uterine and placental tissues from maternal inflammation generated by LPS, reducing the incidence of critical preterm birth [82].

The relevance of melatonin in the placental physiology is further highlighted in preeclampsia, a serious pathological condition affecting pregnant women and characterized by maternal hypertension, proteinuria, placental dysfunction, and often presenting alterations in angiogenic factors [83]. The experimental study of Lanoix et al. compared the expression of melatonin and the levels of its molecular interactors/precursors/receptors, as well as the enzymes for its synthesis, in pre-eclamptic human placentas and control normotensive placentas [84]. The biosynthesis of melatonin requires serotonin acetylation, mediated by aralkyl-amine N-acetyltransferase (AANAT), to produce N-acetyl-serotonin, which is then converted to melatonin by acetyl-serotonin methyltransferase (ASMT) [85].

A reduced AANAT gene expression and activity, along with a strongly reduced ASMT activity, feature the preeclamptic placentas with respect to the controls. In the same way, low levels of melatonin and both MT1 and MT2 receptors characterize pre-eclamptic placentas, while serotonin levels are increased. These findings suggested that the decreased levels of melatonin associated with preeclampsia derive from incorrect AANAT enzyme functioning, which impairs the indolamine production [16]. On these bases, treatment with melatonin was suggested to counteract preeclampsia occurrence in pregnant women (Figure 2) [16]. Moreover, melatonin was deeply investigated for its beneficial antioxidant properties on the fetal development and for the positive impact on IVF-ET techniques, when used as supplement for the embryo culture media [86]. Indeed, melatonin protects the fetus from the oxidative/nitrosative stress by scavenging free radicals, favoring the activation of antioxidant enzymes, and preventing the mitochondrial mediated apoptosis. This activity proved to be important in the treatment of the oocyte donors and during the in-vitro culture, resulting in a strong improvement of the embryo quality [86].

As recent studies pointed out that melatonin plays an important role in the embryo development and in the correct transfer of the molecular information from the mother to the child, the increasing myo-ins requirement during the blastocyst formation and implantation suggests a potentially analogous molecular handover (Figure 2). Such trend of increasing myo-ins concentration in the developing embryos suggests its importance in sustaining fetus growth [87]. In the embryo, myo-ins covers multiple functions as it is the precursor for cell membrane components, promotes the expansion of amniotic and coelomic cavities, and supports cellular metabolism, including the nucleic acid synthesis by providing substrates for the pentose phosphate pathway [87]. Interestingly, in the early weeks of human pregnancy (5–12) myo-ins is significantly higher in the embryonic structures than in maternal serum, as confirmed from the examination of human intervillous, coelomic, and amniotic fluids collected over advancing gestational period. These findings suggested an active carrier-mediated transport mechanism of inositol through the placenta (which builds up in early in pregnancy), later followed by placental/fetal inositol production (Figure 2) [88]. In a recent work, Yang and colleagues highlighted the importance of myo-ins passage from mother to child. Using mouse model knockout for IP3 receptors, they demonstrated that an altered inositol pathway is linked to embryonic mortality and allantoic-placental defects, indicating the molecular involvement of myo-ins in the fetal-maternal connection and embryonic viability [89].

The yolk sac seems responsible for mother-to-fetus inositol delivery. Indeed, in the first trimester, maternal blood flows through intercellular spaces, without directly interacting with the placenta. Through the intervillous space surrounding the placenta, the blood arrives to the yolk sac, which is connected with the embryonic gut and vitelline circulation, thus allowing the transport of nutrients from maternal serum to the embryo [90]. The high demand for myo-ins in the early pregnancy can be also observed in rodents. Experiments demonstrate that myo-ins availability is essential in the developing embryo during pregnancy, when the neural tube shaping occurs. Indeed, inositol deficiency in cultured rodent embryos during neural tube closure led to neural tube defects (NTDs). In this context, inositol supplementation represents a crucial way to reduce embryopathies, including NTDs associated with hyperglycemia, diabetes, and folate deficiency [91].

The observations in mouse models encouraged researchers to further investigate the potential benefits of myo-ins in clinical practice and its involvement in several pathologies associated with human pregnancy. Myo-ins, being second messengers of insulin [92], exhibit an insulin sensitizing action, which explains the therapeutic use of this molecule in patients affected by insulin resistance and diabetes [93,94]. According to recent evidence, congenital malformations affecting the central nervous system seem to occur with higher frequency in diabetic women with respect to the general population, suggesting that inositol presence may be critical in the physiological closure of the neural tube [95]. The wide involvement of myo-ins in the reproductive functions prompted several clinical studies to elucidate the effectiveness of maternal supplementation in the periconceptional period in reducing the NTD risk associated with phenotypes like spina bifida or anencephaly (Figure 2) [91]. Starting from the evidence that maternal diet plays an important role in influencing NTD prevalence [96], clinical observations highlighted that myo-ins may support and enhance the treatment with FA normally used to prevent NTDs. At the same time, myo-ins treatment represents an alternative solution in folate resistant patients [93]. Interestingly, myo-ins supplement in pregnancy seems to reduce NTD occurrence in women who had previous pregnancies with NTDs [97].

In this regard, a clinical trial carried out in Italy and the United Kingdom investigated the benefits of myo-ins supplementation in high-risk women, with 1 or 2 previous pregnancies affected by NTDs. Italian women were treated with 500–1000 mg/day myo-ins plus 5 mg/day FA, starting 2 months pre-conception until 60 days after pregnancy confirmation. Among 29 pregnancies in 27 women no NTD was observed, although about 2–8 NTD cases would be expected from a statistical inference based on occurrence frequency. Most of the women were treated with FA in previous NTD-affected pregnancies, thus proving to be folate-resistant [97,98]. A phase I/II double-blind case-control clinical trial was instead carried out on British women with previous NTD gestation, who desired another pregnancy. Out of 117 contacted women, 99 were eligible and 47 filled a detailed screening questionnaire, agreeing to be randomized to periconceptional supplementation with myo-ins (1 g/day) plus FA (5 mg/day), or only FA as control. Among the 33 randomized pregnancies observed, the authors reported one NTD case in the control group (*n* = 19) and no NTDs in women treated with myo-ins plus FA (*n* = 14). Additionally, 52 women declined randomization, achieving a total of 22 pregnancies. Two NTD episodes occurred in the group under treatment with FA (*n* = 3), while no episodes of NTDs occurred in women under myo-ins plus FA treatment (*n* = 19) [99]. These results are quite remarkable, considering a 3% recurrence risk after one NTD pregnancy and 10% after two [100], supporting a preventive effect of myo-ins on NTD recurrence in high-risk pregnancies.

Moreover, the physiological increase of inositol in the growing embryo is also connected with a major activity of myo-ins, namely the second messenger of insulin signaling [101]. Indeed, maternal insulin status can widely vary during pregnancy, causing deregulation in the maternal-fetal crosstalk. Such abnormal insulin production and insulin resistance phenomena are often observed in pregnancy. The deregulation of insulin status and the concomitant high glucose level in blood lead to a pathological condition named Gestational Diabetes Mellitus (GDM) [100,102]. GDM is normally associated with several risks for the developing embryo, as high glucose levels in the placenta expose the fetus to an unsafe environment, potentially leading to miscarriages. By inhibiting the sodium-myo-inositol transporter 1 (SMIT1) and the inositol monophosphatase 1 (IMPA1), elevated glucose levels hamper myo-ins transport and synthesis, consequently deregulating myo-ins pathway in the fetus [103]. Usually, GDM is associated with familiar diabetes or overweight condition, as in PCOS cases. In this context, myo-ins plays an effective role in the restoring metabolic parameters [104]. Data from clinical trials indicate that myo-ins can prevent GDM occurrence in PCOS women [105,106]. The beneficial effect of myo-ins in reducing GDM occurrence was also demonstrated in women with different types of risk, including parents with type-2 diabetes mellitus, obesity, overweight condition. The study proved that participants receiving 2 g of myo-ins two times daily, from the third trimester of gestation to term, exhibit a significant 60% decrease in GDM occurrence with respect to controls [107].

The mentioned experimental results confirm the positive impact of melatonin and myo-ins supplementation in several pathological and pre-pathological contexts connected to pregnancy, strongly supporting the correct fetal development (Figure 2).

## 6. Childbirth

The information transfer of the circadian rhythm driven by melatonin from mother to child during gestation is completed with the childbirth. Melatonin receptor 1B (MT2, also referred to as hMTNR1B) seems to be directly involved, in synergy with the oxytocin activity, in the promotion of nocturnal uterine contractions. Particularly, in the last phases of gestation melatonin regulates the timing and the degree of contractions. On the contrary, daylight inhibition of melatonin release reduces myometrial contractility [79]. Data demonstrate that myometrial hMTNR1B is silenced during pregnancy and activated towards term to promote contractile functions, in concert with oxytocin receptors. Activation of hMTNR1B induces the transcription of connexin-43 through a protein kinase C (PKC)-dependent mechanism, which leads to increased cell-to-cell coupling signal. Experimental results indicate that the overexpression of circadian locomotor output cycles kaput/brain and muscle ARNT-like 1 (BMAL1/CLOCK) complex causes a rhythm expression of hMTNR1B, thus demonstrating that hMTNR1B is under circadian control [108].

Experiments on pinealectomy (PINX) model in rats indicate that exogenous administration of melatonin can restore the daytime birth pattern observed in the wild type rats. These data also demonstrated that melatonin secretion synchronized with the photoperiodic rhythm is fundamental to determine the parturition time in pregnant rats. Moreover, the increased melatonin levels detected before parturition may indicate the necessity of a circadian signal to induce the delivery [70]. Interestingly, human urinary excretions of melatonin at birth in full-term infants are higher than in pre-term babies. Moreover, pre-term infants require around 8–9 months to achieve the same increase in melatonin levels that full-term children achieve in the first 3 months of life. Analogous delay is observed to reach the rhythmic secretion of melatonin by the pineal gland [109].

Van Dalum and colleagues postulated the existence of a maternal photoperiodic program (MPP), established via melatonin signal, that is transmitted to the fetus. The observation raises from the discovery of melatonin binding sites in rodent, consistently found in pars tuberalis (PT) and pars distalis (PD) of the fetus pituitary and SCN. Melatonin receptors persist after birth in PT, seemingly pivotal for the seasonal actions of melatonin in adult mammals. PT possesses the highest number of melatonin receptors among the mammalian tissues, acting as a circadian time controller and modulating the production of thyrotropin stimulating hormone (TSH) in the thyroid [110].

Further evidence from animal models indicates a role of melatonin in the neuroprotection from perinatal brain injury, and in the protection from brain damages linked with prematurity or birth asphyxia, preventing the associated long-lasting adverse events [111]. Indeed, melatonin reduces hypoxia-ischemia risk and protects immature rat brain from oxidative damage [112]. A clinical study described such neuroprotective activity also in human physiology, observing a decreased serum concentration of oxidative stress markers in asphyxiated newborns supplemented with melatonin (Figure 2). An analogous outcome was observed in infants affected by respiratory distress syndrome and bronchopulmonary dysplasia, when treated with melatonin [113].

As previously described, ongoing pregnancy requires a fine molecular tuning in which species like myo-ins are constantly involved to support to the process. A growing amount of data supports the role of myo-ins in ensuring safe childbirth and in promoting positive gestational outcomes at term (Figure 2). As previously mentioned, myo-ins supplementation may prevent GDM onset [114], not only in pregnant PCOS but also in non-PCOS women [115]. Indeed, the administration of myo-ins to non-PCOS women with elevated fasting plasma glucose (FPG) in the first trimester or early second trimester, strongly improved the maternal Oral Glucose Tolerance Test (OGTT), almost erasing the requirement of additional insulin treatments. More importantly, myo-ins counteracts the occurrence of hypoglycemia in the offspring and contributes to extend the gestational age of approximately 2 weeks [115]. Hypoglycemia is a condition that usually occurs in newborns when the gestation is characterized by high maternal and placental insulin levels, causing the decrease of glucose levels in the fetus. Interesting data from a recent randomized controlled trial demonstrate that patients treated with inositol (specifically the 40:1 combination respectively of myo-ins and D-chiro-inositol (D-chiro-ins), an isomer of the inositol family) exhibit a significant reduction in episodes of neonatal hypoglycemia [116]. 

Until childbirth, the placenta is the vehicle of communication between mother and fetus. In the case of maternal diabetes, excessive maternal-fetal glucose transfer through the placenta may cause fetal macrosomia [117]. Pregnant women with GDM give birth to offspring with greater fat mass with respect to non-GDM women, suggesting that in GDM disrupts the correct transfer of nutrients such as lipids [118].

A longitudinal mother-offspring cohort study demonstrated that high myo-ins amounts in the placenta reduce the pro-adipogenic effects of maternal elevated glycaemia in the newborns. The authors also observed a positive association between maternal glycaemia and birthweight associated with abdominal adiposity when the placenta had low inositol levels. At the same time, high placental myo-ins seems to improve maternal FPG, usually associated with adiposity in the offspring. This is a clinically crucial aspect, given that central adiposity, especially the presence of deep subcutaneous adipose tissue (dSAT), is commonly associated with insulin resistance, inflammation, and increased cardiovascular risks in the children. Data suggest that correct amounts of myo-ins in the placenta are necessary for the physiological development of the offspring, and that myo-ins supplementation during gestation may reduce the risk of excessive fetal adiposity and the related long-term effects on metabolic alterations [117].

The recent experimental work of Longo et al. provided further evidence in this regard. The study evaluated the efficacy of myo-ins supplementation in pregnant mice to improve fetal parameters, with metabolic and cardiovascular focus. Using models for pregnant mice with either obesity or metabolic syndrome, the authors observed that maternal supplementation with inositol (myo-ins and D-chiro-ins in the 40:1 physiological blood ratio) resulted in leaner pups with lower glucose levels and lower systolic blood pressure compared to pups from controls. Furthermore, inositol administration improved cardiovascular functioning and muscular response to contractile and vasorelaxants treatments, independently from the genotype [119].

Myo-ins also plays a critical role in development of the respiratory system, and it proved effective in the prevention and treatment of respiratory distress syndrome (RDS) (Figure 2) [120]. Explanations come from preclinical studies on a pregnant mouse model with a knockout for SMIT1, a myo-ins co-transporters. The pups exhibited low levels of myo-ins associated with respiratory insufficiency, irregular breathing, hypoventilation, and apnea, dying a few days after birth. These effects derive from functional abnormalities in the central mechanisms of respiratory control, which seems to improve after myo-ins supplementation, resulting in increased survival percentage [121].

Altered respiratory functions represent the major consequences of preterm birth (<38 weeks gestation) in humans, often associated with immature infants’ poorly developed respiratory system. Respiratory complications as periodic breathing, apnea, and falls in oxygenation/intermittent hypoxemia (IH) are common features in extremely preterm infants. In the 4 weeks after birth, an increasing number of IH events is inversely related with serum myo-ins levels. Whether myo-ins variations are the cause or the consequence of the IH occurrence is still unclear, myo-ins supplementation represents a powerful support for the respiratory functions, preventing apnea and IH events in newborns [122]. Several data highlighted the myo-ins involvement in the maturation of pulmonary surfactant phospholipids, improving the mechanical and biochemical properties of alveoli and enhancing the resistance to collapsing forces [123]. The role of myo-ins in pulmonary protection was investigated in premature infants affected by RDS. Hallman and colleagues observed improved survival with no bronchopulmonary dysplasia occurrence in neonates undergoing inositol supplementation [124]. Furthermore, Howlett et al. in a Cochrane study confirmed the role of myo-ins supplementation in reducing bronchopulmonary dysplasia and RDS phenomena, by promoting biosynthesis and secretion of surfactant phospholipids in newborns lung tissue [125].

## 7. Conclusions

Increasing evidence highlights the fundamental role of melatonin and myo-ins in human reproductive system and in the achievement of a physiological pregnancy. From the early steps of gestation to childbirth, these species serve as molecular support to ensure the correct embryo formation and development.

In seasonal mammals, the variation in melatonin production is used to synchronize neuroendocrine rhythms with the day-cycle. There are several biological processes regulated by melatonin as well several organs provided with melatonin receptors. The ovarian tissue in women is directly subjected to the melatonin signaling. The scavenger activity allows melatonin to ameliorate the oocyte quality by counteracting the oxidative stress. Its correct amount in the follicular fluid is associated to an improved pregnancy rate. Both mouse models and clinical data demonstrate that melatonin is pivotal in the blastocyst formation and embryo growth, reducing the implantation failure. Melatonin ensures the correct hand-over of the circadian rhythm from mother to fetus through the placenta, and it ameliorates the embryo quality.

In addition, myo-ins is a key regulator of follicular microenvironment, inducing oocyte physiological development. Adequate myo-ins quantities are a marker of good oocyte quality and myo-ins administration is effective in improving oogenesis, as demonstrated in PCOS women. Myo-ins favors the meiotic progression of oocytes into fertilization-competent eggs and reduces the quantity of exogenous gonadotropin required for ART techniques. Additionally, myo-ins promotes the zygote development, improving the embryo quality and the implantation rate. The presence of myo-ins sustains the progression of gestation, preventing the occurrence of negative outcomes and the onset of pathology like NTDs in the fetus.

Noteworthy, when melatonin and myo-ins are supplemented together, their synergic effect enhances the gestational outcomes in natural-occurring pregnancies, contributing to physiological development of the embryo (Table 1).

Most importantly, in IVF procedures, melatonin and myo-ins, either administered to the patients or added to embryo culture media, guarantee an improvement of positive outcomes and increase the chances of success in women willing pregnancy. These molecules sustain the gestation advancing and prevent negative condition that could arise from the deregulation of the complex biochemical machinery underpinning pregnancy.

So far, dietary supplementation with melatonin and myo-ins represents a valid approach to sustain positive gestational outcomes. However, further investigations will elucidate the optimal prescription to be used. Furthermore, the molecular mechanism allowing the efficacy of these species are still not fully described and need to be deeply investigated.

## Figures and Tables

**Figure 1 ijms-22-08433-f001:**
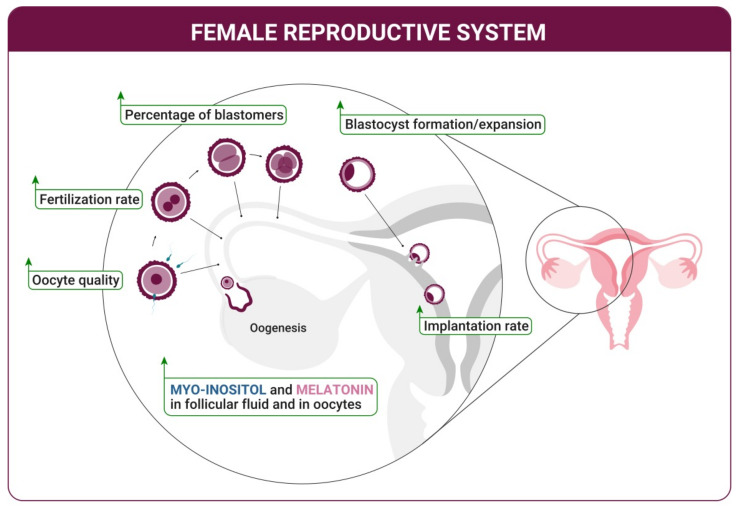
The presence of myo-inositol and melatonin in oocytes and in the follicular fluid positively affects the oogenesis, ameliorates the quality of the oocytes, and promotes the fertilization phase. The result is an increased percentage of blastocyst formation and expansion. Myo-inositol and melatonin sustain the correct attachment of the blastocyst to the uterine endometrium allowing an improvement of the implantation rate.

**Figure 2 ijms-22-08433-f002:**
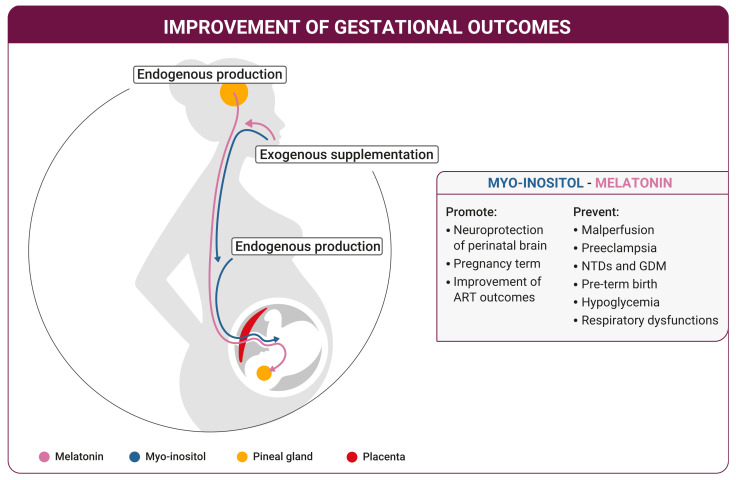
Improvement of gestational outcomes. A dietary supplementation of myo-inositol and melatonin, along with the endogenous production, optimize their circulating levels in the female body improving the gestational outcomes. There is a molecular hand-over of these molecules from the mother to the fetus through the placenta. The melatonin production from the pineal gland of the mother stimulates the pineal gland of the fetus until it becomes able to produce its own melatonin and to self-sustain its own circadian rhythm. Myo-inositol and melatonin promote neuroprotection of the fetus brain, favor the physiological pregnancy term, and improve the ART outcomes, as well as prevent the occurrence of pathological conditions during gestation.

**Table 1 ijms-22-08433-t001:** Summary of melatonin and myo-ins activity during pregnancy.

	Melatonin	Myo-Inositol
Reproductive Apparatus	Higher melatonin level in follicular fluid than in serum	Second messenger of FSH High concentration in ovary
Oogenesis	Reduced oxidative status Increased oocyte quality Increased female fertility Improved women fertility in IVF	Enhanced percentage of MII oocytes Increased oocyte quality Positive outcome in PCOS women Lower rFSH amount in IVF
Oocyte Maturationand and Fertilization	Improved fertilized oocytes Increased early cleaved zygotes Enhanced blastomeres with regular symmetry High-quality day 3 embryos Improvement of the maturation medium in ICSI	Promotion of oocyte maturation Increased fertilization rate Increased percentage of blastomeres forming the embryos Improvement of meiotic progression of oocytes into fertilization-competent eggs Increased number of mature oocytes in ART
Blastocyst Development and Implantation	Enhanced rate of blastocysts and hatching blastocysts development Promotes blastocyst activation and endometrial receptivity Reduced aneuploidy and implantation failure	Enhanced expanded blastocyst stage Increased percentage of 8 and 16-cell embryo Promotion of rapid blastocysts expansion
Fetus Growth	Improved handover of circadian rhythm trough placenta Reduced risk of malperfusion Reduced risk of preeclampsia Improved embryo fetal development in IVF-ET	Higher concentration in embryonic structures than in maternal serum Reduced risk of NTDs Reduced risk of GDM
Childbirth	Regulation of contraction timing Improvement of correct term Neuroprotection of perinatal brain	Reduced hypoglycemia occurrence Reduced risk of fetal macrosomia Prevention of respiratory dysfunctions

## Data Availability

Not applicable.

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
