# Peer review of "Melatonin and Myo-Inositol: Supporting Reproduction from the Oocyte to Birth"

_ijms, 2021, doi:10.3390/ijms22168433_

Round 1

Reviewer 1 Report

The paper by Russo et al. reviewed the main aspects of gestation, embryo implantation, and childbirth highlighting the participation of melatonin and myo-inositol in many physiological and molecular mechanisms to ensure health life to mother and fetus. They concluded that dietary supplementation with melatonin and myo-Ins represents a valid approach to sustain positive gestational outcomes.

Although interesting, this manuscript needs some improvements. There are many typing errors over the manuscript or repeated words. Paper should be carefully revised to improve its readability.

In introduction, it should be stated that “melatonin is secreted daily by the pineal gland during darkness, but also it is produced by almost all cells in a non-circadian manner” (see PMID: 31970423).

The last paragraph of melatonin’s description needs more literature to support the role of melatonin controlling the reproductive functions including their antioxidant functions, hormonal regulation, gamete functionality, like others (see PMID: 24132226, PMID: 24118696, PMID: 21344135, PMID: 31531613).

A general illustrative figure depicting the general effects of melatonin and myo-Ins in early steps of gestation, blastocyst attachment, delivery and childbirth would favor the understanding by the readers.

In conclusion, authors stated that “dietary supplementation with melatonin and myo-Ins represents a valid approach to sustain positive gestational outcomes”. But this should be carefully mentioned since women who are good melatonin/myo-inositol producers would be at higher circulating levels with other unpredictable consequences. The replacement is only recommended to women displaying low mel-myo-ins levels or facing some gestational failures.

Minor comments

Write melatonin and myo-inositol with lower-case letters over the text

Avoid the use of broken paragraphs with little description. These sentences should be connected accordingly.

Author Response

Dear Editor,

We would like to thank you for your helpful comments. We updated the manuscript as suggested. Please find below our response to your comments.

  • Although interesting, this manuscript needs some improvements. There are many typing errors over the manuscript or repeated words. Paper should be carefully revised to improve its readability.

We carefully checked the text for typos and revised the manuscript to improve readability and English form. 

  • In introduction, it should be stated that “melatonin is secreted daily by the pineal gland during darkness, but also it is produced by almost all cells in a non-circadian manner” (see PMID: 31970423).

The non-circadian secretion of melatonin is not mentioned given the focus on the pineal production. In this case we added the mitochondrial secretion for exhaustive information following your suggestion (Lines 58-60).

  • The last paragraph of melatonin’s description needs more literature to support the role of melatonin controlling the reproductive functions including their antioxidant functions, hormonal regulation, gamete functionality, like others (see PMID: 24132226, PMID: 24118696, PMID: 21344135, PMID: 31531613) (Lines 73-74).

We revised the paragraph inserting the other functions recommended and we added the references accordingly.

  • A general illustrative figure depicting the general effects of melatonin and myo-Ins in early steps of gestation, blastocyst attachment, delivery and childbirth would favor the understanding by the readers.

As suggested, we inserted two images in the paper: Figure 1 – line 217, Figure 2 – line 615.

  • In conclusion, authors stated that “dietary supplementation with melatonin and myo-Ins represents a valid approach to sustain positive gestational outcomes”. But this should be carefully mentioned since women who are good melatonin/myo-inositol producers would be at higher circulating levels with other unpredictable consequences. The replacement is only recommended to women displaying low mel-myo-ins levels or facing some gestational failures.

Administration of melatonin and myo-inositol in the cited studies is performed according to the limits of molecules content established for dietary supplementation at the time of the study.

The patients enrolled in the different studies were healthy (PMID: 11068941, PMID: 22823904) as they were not lacking in the circulating levels of melatonin and myo-inositol. Is likely that among the subjects included there were good melatonin/myo-inositol producers. In all the studies were retrieved no adverse effects after melatonin and myo-inositol treatment.

Furthermore, available data regarding the safety of the supplementation with these natural species highlight the absence of risk associated to long term treatments (PMID: 27843451, PMID: 21845803, PMID: 26692007).

Minor comments

  • Write melatonin and myo-inositol with lower-case letters over the text.

Thank you for noticing, we corrected the issue.

  • Avoid the use of broken paragraphs with little description. These sentences should be connected accordingly

We reformatted the main text according to this indication.

We believe that our corrections according to your recommendations improve the manuscript and make it complete and suitable for publication in its current form.

Yours sincerely,

On behalf of all the authors,

Reviewer 2 Report

Authors summarized the melatonin and myo-inositol for supporting reproductioan from the ooctye to birth. Melatonin is recognized as important hormone in seasonal reproduction. Additionally, it has been reported as efficient supplement in preimplantation development of in vitro embryos. This reveiw paper is well summarized about melatonin and myo-inositol and I recommend the 1 graphical abstract and 2 figures for more reading by attraction from researchers in this field. In my thought, this paper would not be cited if it is published without any figures.  

Author Response

We thank the reviewer for their valuable comments.

As suggested we inserted two images in the paper: Figure 1 – line 217, Figure 2 – line 615. 

Round 2

Reviewer 1 Report

No additional comments